# The Effect of Pre-Exercise Caffeine and Glucose Ingestion on Endurance Capacity in Hypoxia: A Double-Blind Crossover Trial

**DOI:** 10.3390/nu16213624

**Published:** 2024-10-25

**Authors:** Chih-Hui Chiu, Chung-Chih Chen, Ajmol Ali, Shey-Lin Wu, Ching-Lin Wu

**Affiliations:** 1Graduate Program in Department of Exercise Health Science, National Taiwan University of Sport, Taichung 404401, Taiwan; loveshalom@hotmail.com; 2Graduate Institute of Sports and Health Management, National Chung Hsing University, Taichung 402202, Taiwan; nike53reebok@hotmail.com; 3School of Sport, Exercise and Nutrition, Massey University, Auckland 0745, New Zealand; a.ali@massey.ac.nz; 4Neurological Department, Show Chwan Memorial Hospital, Changhua 500209, Taiwan; 5Neurological Department, Chang Bing Show Chwan Memorial Hospital, Changhua 500209, Taiwan; 6Faculty of Kinesiology and Physical Education, University of Toronto, Toronto, ON M5S 2W6, Canada

**Keywords:** hypoxia, glycogen, fat oxidation, endurance performance

## Abstract

The impact of caffeine and glucose supplementation in a hypoxic environment on endurance exercise performance remains inconclusive. The current study examined the effect of pre-exercise carbohydrate and caffeine supplementation on endurance exercise performance in an acute hypoxic environment. Eight healthy active young males participated in this double-blind, within-subjects crossover study. Participants ingested the test drink 60 min before exercising at 50% Wmax for 90 min on a cycle ergometer (fatiguing preload); there followed an endurance performance test at 85% Wmax until exhaustion in a hypoxic chamber (~15%O_2_). Participants completed four experimental trials in a randomized order: caffeine (6 mg·kg^−1^; Caff), glucose (1 g·kg^−1^; CHO), caffeine (6 mg·kg^−1^) + glucose (1 g·kg^−1^; Caff−CHO), and taste- and color-matched placebo with no caffeine or CHO (PLA). Blood samples were collected during fasting, pre-exercise, every 30 min throughout the exercise, and immediately after exhaustion. The caffeine and glucose trials significantly enhanced endurance capacity in hypoxic conditions by Caff, 44% (68.8–31.5%, 95% confidence interval), CHO, 31% (44.7–15.6%), and Caff−CHO, 46% (79.1–13.2%). Plasma-free fatty-acid and glycerol concentrations were higher in Caff and PLA than in CHO and Caff−CHO (*p* < 0.05). The estimated rate of fat oxidation was higher in Caff and PLA than in CHO and Caff−CHO (*p* < 0.05). There were no significant differences in ratings of perceived exertion between trials. In conclusion, the ingestion of caffeine, glucose, or caffeine + glucose one hour before exercising in hypoxic conditions significantly improved 85% Wmax endurance performance after prolonged exercise.

## 1. Introduction

Living at sea level while training in intermittent hypoxic conditions has become more commonplace in recent years [1,2]. Exercise in hypoxic conditions decreases pulse oximetry (SPO_2_) and fat oxidation [3] and increases heart rate, blood lactate concentration, and glycogenolysis in exercising muscle [4], which may accelerate muscle glycogen depletion during prolonged exercise [5,6]. Therefore, delaying muscle glycogen depletion in acute hypoxia through the provision of exogenous carbohydrate (CHO) may enhance the training effect, i.e., improve endurance performance and/or enhance fat utilization when competing at sea level [7]. 

It has been suggested that the ingestion of CHO before exercise spares muscle glycogen and improves endurance exercise performance in normoxia conditions. The possible mechanisms include increased exogenous CHO oxidation, resynthesis of muscle glycogen, and/or delayed muscle glycogen depletion [8]. Prolonged moderate-intensity exercise depletes muscle glycogen [9] in normoxia. However, acute hypoxic exposure impairs glucose uptake and reduces the oxidation rate of exogenous CHO intake [10]. In addition, the rate of CHO oxidation increases at the same exercise intensity compared to normoxia [3]. The increment in CHO utilization may hasten muscle glycogen depletion. Therefore, it is important to know whether CHO supplementation decreases muscle glycogen depletion and improves exercise capacity during hypoxia.

Caffeine supplementation before endurance exercise has been suggested to increase fat utilization and delay muscle glycogen depletion during prolonged exercise [11,12]. In a systematic review and meta-analysis, caffeine supplementation before exercise was shown to increase fat oxidation during submaximal endurance exercise [13]. Caffeine supplementation before exercise can also increase plasma free fatty acid (FFA) concentration and increase fat oxidation [14], which may help to decrease the rate of muscle glycogenolysis and delay muscle glycogen depletion. Although caffeine ingestion increases fat lipolysis and fat oxidation, the ergogenic benefits of caffeine intake are more likely due to the antagonism of adenosine receptors, which reduces central nervous system (CNS) fatigue, the increased release of calcium ions from the sarcoplasmic reticulum, and/or the maintenance of sodium–potassium ATPase (Na^+^/K^+^-ATPase) activity [15]. However, there is no evidence on whether caffeine supplementation can improve endurance exercise performance in an acute hypoxic environment. In addition, the co-ingestion of glucose and caffeine increases the absorption rate of glucose [16] and improves muscle glycogen resynthesis. These mechanisms may result in additional improvements in exercise performance compared to the ingestion of glucose or caffeine alone.

There are limited data on CHO supplementation and exercise performance at high altitude [17,18,19,20], and the findings are inconsistent. Two studies show that CHO supplementation at high altitude improves mountaineering [20] and cycling time-trial [19] performance, but other studies show no benefit of CHO supplementation on exercise performance [17,18]. The differences in acute altitude exposure time, daily energy balance, and the doses of CHO supplementation may influence the ergogenic effects of CHO supplementation on exercise performance. Acute exposure to hypoxic environments is a challenge faced by many mountaineers and hikers, and providing nutritional support to assist or maintain their physical performance is an important task for their health and well-being, as well as for their performance. Therefore, more research is warranted to investigate whether CHO supplementation can aid performance in hypoxic conditions/at altitude. In addition, no study has investigated the effect of caffeine supplementation on endurance performance in hypoxia. It was hypothesized that caffeine and CHO supplementation might have a synergistic effect on fat oxidation, maintaining higher glycogen concentrations after endurance exercise in an acute hypoxic environment. The current study examined the effect of CHO and caffeine supplementation on endurance exercise capacity, FFA availability, and fat oxidation in acute hypoxic environments among healthy, active adult males.

## 2. Materials and Methods

### 2.1. Participants

Eight healthy, active, recreationally trained ($\overset{\cdot }{V}\epsilon$O_2_ max: 39.9 ± 4.3 mL·kg^−1^·min^−1^ in hypoxia, Wmax: 203.1 ± 10.0 watt) collegiate students volunteered for this study (age: 21.9 ± 0.8 years, height: 174.5 ± 2.1 cm, weight: 76.1 ± 5.1 kg). Participants were required to complete a health questionnaire before taking part in the study. All volunteers provided written informed consent after explanations of the experimental protocol and the possible risks of the study. All participants were non-habitual caffeine users, and none were under medication. The protocol of the study was approved by the Human Ethical Committee of the National Taiwan University of Sport (NTUS10803).

### 2.2. Sample Size Calculation

The required sample size was determined using G*Power software (version 3.1.9.4, Universität Düsseldorf, Kiel, Germany) [21]. The calculation was conducted with an alpha level of 0.05 and a correlation coefficient of 0.80. Fulco et al. (2005) found that CHO supplementation in an acute hypoxic environment significantly reduced the duration of the time-trial test, with an r-value of 0.91 and a *p*-value of <0.001 [19]. A sample size of 4 was determined to be sufficient for the detection of differences between trials, as indicated by the results of the analysis. A sample size of 8 participants was deemed appropriate for the purposes of elucidating the statistical discrepancies.

### 2.3. Experimental Design

Participants were asked to complete 4 experimental trials for this double-blinded and randomized crossover study in a Latin-square design: (i) caffeine (Caff, 6 mg·kg^−1^); (ii) glucose (CHO, 1 g·kg^−1^); (iii) caffeine and glucose (Caff−CHO); and (iv) placebo (PLA). Participants were required to ingest the beverage 1 h before exercising in a hypoxic chamber (15% O_2_). The exercise was set at 50% Wmax for 90 min on a cycle ergometer and followed by an endurance capacity test set at 85% Wmax until volitional exhaustion. Each trial was separated from the previous one by at least one week.

### 2.4. Protocol

#### 2.4.1. Preliminary Test

Participants were asked to perform a maximal oxygen uptake ($\overset{\cdot }{V}\epsilon$O_2_ max) test on a cycle ergometer using a pre-calibrated breath-by-breath gas analyzer (Cortex, Metamax 3B, Leipzig, Germany) in a hypoxic chamber (Colorado Altitude Training, Boulder, CO, USA) one week before the main trial. The hypoxic chamber was set at 15% O_2_ (simulated ~2300 m altitude). Participants were asked to sit in the hypoxic chamber 30 min before the test. The initial load was set at 75 W and increased by 25 W every 3 min until volitional fatigue. 

#### 2.4.2. Main Trial

Participants reported to the laboratory at 08:00 in the morning after an overnight fast. After anthropometric measurement, a cannula was inserted into a forearm vein; blood samples were collected at fasting and at 30, 60, and 90 min during exercise and at exhaustion. Participants were then asked to ingest one of the four test drinks after the fasting blood sample was taken. Participants then rested for 60 min before the exercise performance test. Thirty min before exercise, participants were asked to sit in the hypoxic chamber (15% O_2_, 2300 m altitude). The exercise was set at 50% Wmax for 90 min on a cycle ergometer. The expired air samples were collected at 30, 60, and 90 min during exercise to calculate the rate of fat and CHO oxidation [22]. The exercise intensity then increased to 85% Wmax until exhaustion, with the duration of exercise recorded as the performance time. 

Participants were asked to avoid heavy physical activities and refrain from any foods or drinks containing caffeine and alcohol 48 h before the main trials. Participants were also required to record their diet for two days before the first main trial by taking photographs and were asked to repeat the same diet in the following trials. The study flow diagram and exercise protocol were as shown in Figure 1.

### 2.5. Test Drink

After the fasting blood sample was taken, participants were asked to consume a 500 mL grape-flavor drink with either a caffeine capsule (6 mg·kg^−1^), glucose (1 g·kg^−1^), a caffeine capsule plus glucose (6 mg·kg^−1^ of caffeine; 1 g·kg^−1^ of glucose), or a placebo. In the CHO and PLA trials, the same mass of flour in a capsule was provided to participants. Glucose was dissolved in the grape-flavor drink in the CHO and Caff−CHO trials. The safety and efficacy of the caffeine doses have been documented in the International Society of Sports Nutrition position stand [23], and these dosages have been used in several studies on normoxia. The dose applied in the study was within the safety range.

### 2.6. Blood Sample Collection

Ten milliliter venous blood samples were collected via a cannula (Venflon 20G, Stockholm, Sweden) connected to a 3-way stopcock (Connecta Ltd., Stockholm, Sweden) with a 10-cm extension tube at each collection time. The blood samples were collected in EDTA tubes then centrifuged at 1500× *g* (Eppendorf 5810, Hamburg, Germany) for 20 min to extract plasma samples. The plasma was aliquoted and stored at −80 °C before analysis.

### 2.7. Blood Analysis

Plasma concentrations of glucose, lactate, non-esterified free fatty acids (FFA), and glycerol were measured using an automated analyzer (Hitachi 7020, Tokyo, Japan) using commercial kits (Randox, Antrim, UK). Plasma concentrations of insulin were measured by performing electrochemiluminescence assays with an analyzer (Elecsys 2010, Roche Diagnostics, Basel, Switzerland) using a commercial kit (Roche Diagnostics, Basel, Switzerland).

### 2.8. Statistical Analysis

All results are presented as means ± SEM. The Shapiro–Wilk test was used to assess the normality of the data distribution. Plasma FFA, glycerol, lactate, and glucose were compared among trials and over time using a two-way analysis of variance (ANOVA) with repeated measures. The trapezium rule was used to calculated the 90 min total area under the CHO and fat oxidation rate versus time curve (AUC). Exercise performance and AUC were compared among trials using a one-way ANOVA with repeated measures. Where significant results were found by ANOVA, post hoc pairwise t-tests were used, using the Bonferroni correction, to assess differences between specific groups. Statistical significance was set at *p* < 0.05.

## 3. Results

### 3.1. Exercise Performance

The endurance performance time was higher in Caff (433.8 ± 85.1 s), CHO (377.9 ± 100.0 s), and Caff−CHO (539.4 ± 94.0 s) than in PLA (245.5 ± 60.4 s) (*p* = 0.038; η^2^ = 0.652; actual power = 0.83) (Table 1). 

### 3.2. Plasma Concentrations of FFA, Glycerol, Lactate, Glucose, and Insulin

Plasma FFA concentrations (Figure 2A) were higher in Caff and PLA than in CHO and Caff−CHO (treatment * time, *p* < 0.001; treatment, *p* < 0.001; time, *p* < 0.001; η^2^ = 0.652). Specifically, significant differences were found between Caff and PLA relative to Caff−CHO and CHO trials at 0 min, 90 min, and exhaustion (*p* < 0.001). Plasma glycerol concentrations (Figure 2B) were higher in Caff and PLA than in CHO and Caff−CHO (treatment * time, *p* = 0.001; treatment, *p* = 0.001; time, *p* < 0.001; η^2^ = 0.565). Specific differences were found between Caff and PLA relative to Caff−CHO and CHO trials at 0, 30 min, and exhaustion (*p* < 0.05). There were no differences among trials for plasma glucose concentrations (Figure 2C) (treatment * time, *p* = 0.092; treatment, *p* = 0.055; time, *p* < 0.001; η^2^ = 0.551). Plasma insulin concentrations (Figure 2D) were lower in Caff and PLA than CHO and Caff−CHO trials, with no difference between the Caff and PLA trials (treatment * time, *p* = 0.035; treatment, *p* = 0.022; time, *p* = 0.002; η^2^ = 0.551). 

### 3.3. Fat Oxidation 

The total rate of fat oxidation (Figure 3A) was higher in the Caff and PLA trials than in the CHO and Caff−CHO trials (treatment * time, *p* = 0.607; treatment, *p* = 0.049; time, *p* = 0.047; η^2^ = 0.198) during the 90-min exercise at 50% Wmax. There was no significant difference between Caff and Caff−CHO or CHO and PLA trials (*p* > 0.05). Rates of fat oxidation during exercise were higher in Caff and PLA than in CHO and Caff−CHO (*p* = 0.005, η^2^ = 0.451; Figure 3B).

## 4. Discussion

The primary finding was that the ingestion of caffeine, glucose, or caffeine + glucose one hour before exercise significantly improved endurance performance in recreationally active males after prolonged exercise in acute hypoxia conditions. The ingestion of glucose before exercise improved endurance performance in acute hypoxia conditions. The co-ingestion of glucose with caffeine did not show an additional increase compared to Caff−CHO on endurance performance in acute hypoxia.

To the best of our knowledge, no study has investigated the effect of CHO supplementation on endurance exercise performance in acute hypoxia; therefore, this is the first study that shows that glucose ingestion improves endurance capacity during acute hypoxia. Exercise during hypoxia results in a higher muscle glycogen utilization rate and might lead to early muscle glycogen depletion [5,6]. Therefore, delaying muscle glycogen depletion is of critical importance for improving endurance exercise performance. In normoxic conditions, CHO supplementation before exercise may improve endurance performance by increasing exogenous CHO oxidation and delaying muscle glycogen depletion. Our data show that the ingestion of glucose before exercise improves the endurance performance under 85% Wmax intensity after 90 min of exercise in hypoxia. A similar magnitude of change in time-to-exhaustion performance (124%) was found in a previous study [24]. The result is similar to studies in normoxic conditions, and the same mechanism may influence the result in hypoxia. 

We also found that the ingestion of caffeine before exercise increased endurance performance in acute hypoxia conditions relative to placebo intake. The possible benefit for caffeine supplementation on endurance exercise performance might be due to the amelioration of painful feelings before exhaustion [25] and/or an increased rate of fat oxidation [26]. In our study, there was no difference in rates of fat oxidation between the CHO and PLA trials. Therefore, the rate of fat oxidation may not play an important role during hypoxia; this supports the findings of other studies undertaken during normoxia [27]. In addition, the Caff and Caff−CHO trials showed 44% and 46% improvements compared to PLA, respectively. We think the central fatigue or decreased feelings of pain may have been an important mechanism in this study when caffeine was ingested.

The co-ingestion of glucose with caffeine did not show an additional increase compared to Caff and CHO trials in endurance performance in acute hypoxia. The co-ingestion of glucose and caffeine has been suggested as a means to improve muscle glycogen resynthesis and increase exogenous CHO oxidation [16], which may decrease endogenous CHO oxidation and delay muscle glycogen depletion. However, others have shown that the co-ingestion of glucose and caffeine did not provide additional changes to substrate utilization compared to CHO ingestion alone [28]. In contrast to these studies, which were completed in normoxic condition, we failed to find any difference between CHO and Caff−CHO trials in plasma glucose, FFA, glycerol concentrations, and fat oxidation rates. From these data, the present study disproved the hypothesis of those studies and found that CHO coupled with caffeine intake did not cause a synergistic effect leading to better endurance exercise performance. Our data suggest that in acute hypoxia, the co-ingestion of caffeine and glucose do not influence exercise metabolism and endurance performance compared to the ingestion of glucose alone. Further research is needed to explore potential differences between findings at normoxia and hypoxia with regard to the individual and additive effects of CHO and caffeine intake on endurance performance. 

The strength of this study is that it is the first to investigate the effectiveness of glucose and caffeine supplementation to improve endurance exercise performance in an acute hypoxic environment. The major limitation of the present study is that we did not measure muscle glycogen concentrations. However, it has been suggested that 90 min of exercise can significantly decrease muscle glycogen concentration. Exercise during hypoxia further increases CHO oxidation. Therefore, we believe that the present study successfully decreased muscle glycogen concentration to different levels, because the exercise exhaustion times after 90 min of exercise were significantly different. We allowed participants to record their diets for the two days prior to the study using photographs and did not provide a standard diet or control sleep duration and hydration states. Although participants were asked to consume the same food at the same point in time two days before the next trial, there may still have been a large margin of error compared to the provision of a standard diet. Subsequent studies will provide a standardised diet in order to optimize the control of external variables prior to the commencement of the formal trial.

## 5. Conclusions

In summary, the ingestion of caffeine, glucose, or caffeine + glucose one hour before exercise significantly improved 85% Wmax endurance performance after prolonged exercise in acute hypoxia conditions (relative to a placebo). The ingestion of either glucose or caffeine alone before exercise improves endurance performance in acute hypoxia conditions. The co-ingestion of glucose with caffeine did not show an additional increase compared to Caff and CHO in endurance performance in acute hypoxia conditions.

## Figures and Tables

**Figure 1 nutrients-16-03624-f001:**
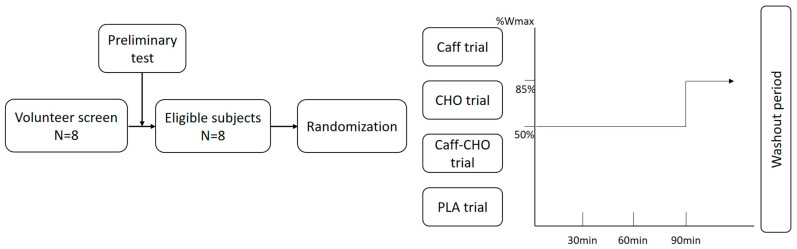
Study flow diagram and exercise protocol. CHO trial: glucose trial; Caff−CHO trial: caffeine + glucose trial; PLA trial: placebo trial.

**Figure 2 nutrients-16-03624-f002:**
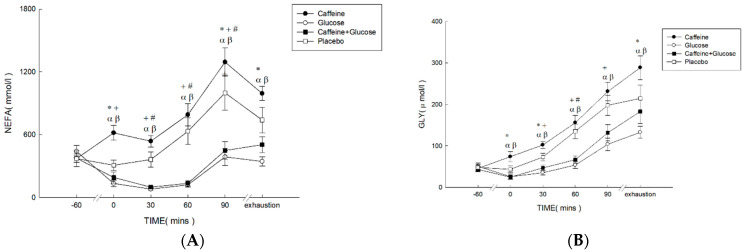
Plasma concentrations of FFA (**A**), glycerol (**B**), glucose (**C**), and insulin (**D**) in the caffeine (Caff), glucose (CHO), caffeine + glucose (Caff−CHO), and placebo (PLA) trials. Values are mean ± SEM, *n* = 8. * Caff trial significantly different from PLA (*p* < 0.05). + CHO trial significantly different from PLA (*p* < 0.05). # Caff−CHO trial significantly different from PLA (*p* < 0.05). α CHO trial significantly different from Caff (*p* < 0.05). β Caff−CHO trial significantly different from C (*p* < 0.05).

**Figure 3 nutrients-16-03624-f003:**
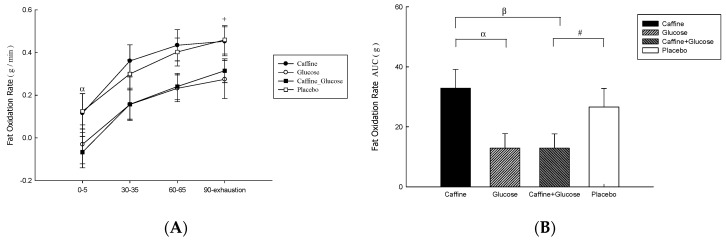
Fat oxidation (**A**) and area under the fat oxidation rate during exercise (**B**) on the caffeine (Caff), glucose (CHO), caffeine + glucose (Caff−CHO) and placebo (PLA) trials. Values are mean ± SEM, *n* = 8. + CHO trial significantly different from *p* (*p* < 0.05). # Caff + CHO trial significantly different from PLA (*p* < 0.05). α CHO trial significantly different from Caff (*p* < 0.05). β Caff + CHO trial significantly different from Caff (*p* < 0.05).

**Table 1 nutrients-16-03624-t001:** The endurance performance time on the caffeine (Caff), glucose (CHO), caffeine + glucose (Caff−CHO), and placebo (PLA) trials.

**Caff**	**CHO**
**433.8 ± 85.1 s ***	**377.9 ± 100.0 s ***
**44.0 ± 23.9% (22.7–65.3%)**	**31.1 ± 13.8% (18.6–43.3%)**
**Caff−CHO**	**PLA** 245.5 ± 60.4 s
539.4 ± 94.0 s *	
**46.2 ± 32.9%** (13.2–79.1%)	

Values are mean ± SEM, *n* = 8. Percentage values show mean (and range) percent differences to PLA. * Significantly different from PLA (*p* < 0.05).

## Data Availability

The original contributions presented in the study are included in the article, further inquiries can be directed to the corresponding author. The data are not publicly available due to privacy or ethical restrictions.

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
