# Peer review of "The Effect of Pre-Exercise Caffeine and Glucose Ingestion on Endurance Capacity in Hypoxia: A Double-Blind Crossover Trial"

_nutrients, 2024, doi:10.3390/nu16213624_

Round 1
Reviewer 1 Report
Comments and Suggestions for Authors
Dear authors, the manuscript is generally well-written and easy to read; a slight spell-check is required. I have just some concerns that the authors must address.
Abstract
I suggest adding a brief background.
What’s the gap you’re trying to fill with your work?
Introduction
The literature on the subject is sufficiently well summarised. However, it could be useful to add some information about:
- You should clearly distinguish between how carbohydrate metabolism might differ between these two conditions (normoxic and hypoxic). There should be a clearer explanation of how hypoxia specifically influences muscle glycogen depletion compared to normoxia, as these environments have distinct physiological effects.
- You highlight the potential benefits of caffeine on fat oxidation and muscle glycogen sparing in normoxia. However, you should consider the possibility that caffeine's stimulatory effects on the CNS may influence exercise performance independently of its effects on fat metabolism. Caffeine's role in increasing fat oxidation is mentioned, but its other mechanisms (e.g., improving alertness and perceived exertion) are not discussed, which is an incomplete assessment of how caffeine could impact performance in hypoxia.
- The direct link between caffeine, fat oxidation, and glycogen sparing sounds speculative, and it’s not adequately supported with mechanistic evidence. Simply increasing fat oxidation does not necessarily translate into significant glycogen sparing, particularly in a hypoxic environment where carbohydrate metabolism tends to dominate.
- It’s unclear how caffeine would function differently in hypoxia compared to normoxia, as there may be specific effects of hypoxia (e.g., reduced oxygen availability, altered metabolism) that change how caffeine affects the body.
- Hypothesis is not entirely clear, are you hypothesizing that both carbohydrate and caffeine will have an additive or synergistic effect on performance in hypoxia?
Methods
- The sample size used looks very small. Have you performed an a priori statistical power analysis?
- Participants were asked to record their diet two days before the first main trial and replicate it in subsequent trials, this method is inherently unreliable because dietary recall and adherence to dietary replication can be inconsistent. A more controlled dietary approach (e.g., providing standardized meals) could ensure greater consistency across trials. At least consider stating it as a limitation.
- You state that each trial was separated at least one week. It means that not all the trial was separated by the same time?
- Placebo drink, as well as the caffeinated and glucose-containing beverages, were identical in taste, appearance, and texture? You mention a grape-flavored drink, but whether participants could detect differences between the caffeine, glucose, and placebo conditions is not clear. If the drinks are distinguishable, this could compromise the blinding and affect the placebo effect.
Results
I could be wrong but consider this:
- The percentage differences between conditions and placebo are confusing or possibly incorrect. I.e.: Caff (215.4%) is reported to have a 132.9%–297.9% range compared to PLA. CHO-Caff (270.7%) seems to have an illogical range of 475.8%–66.6%. The range should be within the possible values for percent improvement over PLA.
Discussion and conclusions
- It could be useful clarify why there was no significant improvement when glucose and caffeine were ingested together in the hypoxia condition.
- Without measuring muscle glycogen levels, conclusions related to glycogen depletion are very speculative.
- Other factors like hydration or sleep could influence performance outcomes, especially in hypoxia where the body is under more physiological stress.
- There is no mention of individual differences in response to caffeine or glucose intake, which can vary widely. Addressing this variability could help explain why some participants may have shown better endurance than others, or why certain trials showed significant results while others did not. Especially considering the smallness of the sample used.
Comments on the Quality of English Language
Minor editing of English language required.
Author Response
Response to Reviewer 1 Comments
Dear authors, the manuscript is generally well-written and easy to read; a slight spell-check is required. I have just some concerns that the authors must address.
RE: Thank you very much for taking the time to review our article. We hope these revisions are up to the journal’s publication standards. We believe that the updated manuscript has substantially improved with the reviewers’ comments and hope it will now be suitable for publication. Many thanks for your kind consideration.
Abstract
I suggest adding a brief background.
What’s the gap you’re trying to fill with your work?
RE: Thank you for your suggestions. We have added a brief background in the abstract (lines 20-21).
Introduction
The literature on the subject is sufficiently well summarised. However, it could be useful to add some information about:
- You should clearly distinguish between how carbohydrate metabolism might differ between these two conditions (normoxic and hypoxic). There should be a clearer explanation of how hypoxia specifically influences muscle glycogen depletion compared to normoxia, as these environments have distinct physiological effects.
RE: Thanks for the comments. We have added the suggested information in the introduction (lines 54-57).
- You highlight the potential benefits of caffeine on fat oxidation and muscle glycogen sparing in normoxia. However, you should consider the possibility that caffeine's stimulatory effects on the CNS may influence exercise performance independently of its effects on fat metabolism. Caffeine's role in increasing fat oxidation is mentioned, but its other mechanisms (e.g., improving alertness and perceived exertion) are not discussed, which is an incomplete assessment of how caffeine could impact performance in hypoxia.
RE: It is insightful suggestion. We are aware of the CNS influence on exercise performance. The information has added in the introduction (lines 66-70).
- The direct link between caffeine, fat oxidation, and glycogen sparing sounds speculative, and it’s not adequately supported with mechanistic evidence. Simply increasing fat oxidation does not necessarily translate into significant glycogen sparing, particularly in a hypoxic environment where carbohydrate metabolism tends to dominate.
RE: Thanks for the comments. We did not include the measurement of muscle glycogen concentrations before/after exercise which is one of the limitations of the study. Indeed, we agree that carbohydrate tends to dominate energy provision during exercise in a hypoxic environment. However, during the same intensity/duration of prolonged endurance exercise in a hypoxic environment, the energy expenditure was assumed similar. If the rate of fat oxidation increased, it is reasonable to speculate that it may spare carbohydrate stores.
- It’s unclear how caffeine would function differently in hypoxia compared to normoxia, as there may be specific effects of hypoxia (e.g., reduced oxygen availability, altered metabolism) that change how caffeine affects the body.
RE: Thanks for your comments. We agree that in a hypoxic environment, the effects of caffeine may differ from those in a normoxic environment. However, to the best of our knowledge, there is limited literature on the effect of caffeine intake in hypoxia. Therefore, the current study design is also based on the potential effects observed in a normoxic environment.
- Hypothesis is not entirely clear, are you hypothesizing that both carbohydrate and caffeine will have an additive or synergistic effect on performance in hypoxia?
RE: Thank you for your suggestions. We have added this information to the introduction (lines 88-90).
Methods
- The sample size used looks very small. Have you performed an a priori statistical power analysis?
RE: Thanks for the comments. We included the calculation of the number of participants in paragraph 2.1 and found that the number of participants in this study should be adequate (line 102-111). Furthermore, an actual power of 0.83 was calculated for the exercise performance data, indicating that the number of participants is sufficient for the experiment. We have added more information on sample size calculation in the revised manuscript (lines 104-112).
- Participants were asked to record their diet two days before the first main trial and replicate it in subsequent trials, this method is inherently unreliable because dietary recall and adherence to dietary replication can be inconsistent. A more controlled dietary approach (e.g., providing standardized meals) could ensure greater consistency across trials. At least consider stating it as a limitation.
RE: Thanks for the comments. The lack of a standard diet may be one of the limitations of the study. We have put the statement in the discussion (lines 284-290).
- You state that each trial was separated at least one week. It means that not all the trial was separated by the same time?
RE: Due to the different lifestyles of the participants, it was difficult to conduct the trial at the same time each week. The washout period was 7-10 days for all participants. All subjects had identical wash-out period between each trial. In addition, we instructed participants to refrain from heavy physical activity for three days prior to the trials and to follow the same diet to ensure optimal control of external variables.
- Placebo drink, as well as the caffeinated and glucose-containing beverages, were identical in taste, appearance, and texture? You mention a grape-flavored drink, but whether participants could detect differences between the caffeine, glucose, and placebo conditions is not clear. If the drinks are distinguishable, this could compromise the blinding and affect the placebo effect.
RE: The flavor and appearance of the glucose and placebo drinks were comparable, as were the shape, appearance, and color of the capsules. Therefore, we concluded that the participants were unable to distinguish between the supplements.
Results
I could be wrong but consider this:
- The percentage differences between conditions and placebo are confusing or possibly incorrect. I.e.: Caff (215.4%) is reported to have a 132.9%–297.9% range compared to PLA. CHO-Caff (270.7%) seems to have an illogical range of 475.8%–66.6%. The range should be within the possible values for percent improvement over PLA.
RE: We apologize for the typing errors. We have re-checked and corrected them (Table 1).
Discussion and conclusion
- It could be useful clarify why there was no significant improvement when glucose and caffeine were ingested together in the hypoxia condition.
RE: Thank you for raising this point. The current data suggests that there was no difference in biochemical parameters between the co-ingestion of CHO and caffeine compared to CHO alone. Therefore, the addition of caffeine to glucose supplementation did not show a synergistic effect on energy metabolism and subsequent exercise performance. We have addressed this in the Discussion section (lines 269-271).
- Without measuring muscle glycogen levels, conclusions related to glycogen depletion are very speculative.’
RE: Indeed, we have made the amendments in conclusion section (lines 292-297).
- Other factors like hydration or sleep could influence performance outcomes, especially in hypoxia where the body is under more physiological stress.
RE: Thanks for the comments. All participants were asked to maintain their lifestyle and diet as consistently as possible during the experiment period. It was still difficult to standardize all external variables as mentioned like sleep or other physiological and psychological stress. We have added these aspects in the limitations of the research section (lines 284-290).
- There is no mention of individual differences in response to caffeine or glucose intake, which can vary widely. Addressing this variability could help explain why some participants may have shown better endurance than others, or why certain trials showed significant results while others did not. Especially considering the smallness of the sample used.
RE: Thank you for the helpful comments. We agree with that there will be responders and non-responders and also inter and intra-individual variation. However, the current data showed that the 95% CI was recalculated for the exercise performance values, revealing that participants exhibited improved in endurance exercise performance compared to the PLA trial, despite the presence of individual differences.

Reviewer 2 Report
Comments and Suggestions for Authors
This is a double-blind crossover trial assessing the the effect of pre-exercise caffeine and glucose ingestion on endurance capacity in hypoxia. My comments are listed below:
1. Introduction: authors must provide additional description on the target population (mountaineers etc.) for this research trial. This research caters to a very specific target population so enough background on this population must be included in this section.
2. Suggest authors to include a flow diagram/figure that provides readers with a quick outline of the research design including a brief description of the activities done at each visit.
3. Results: Figure 1 is not mentioned the text and I feel Figure 1 can be presented as a Table.
3. Discussion: authors must expand upon the value/significance of their research including details on the possible mechanism(s) of action(s). As currently written this section does not provide sufficient discussion of study results.
General comments:
- please elaborate abbreviations at the first use in the manuscript.
- This manuscript needs professional English language assistance.
- Authors should include a copy of their informed consent document as a supplemental file.
Comments on the Quality of English Language
Poor English with plenty grammatical mistakes, This manuscript needs professional English language assistance.
Author Response
Response to Reviewer 2 Comments
Thank you very much for taking the time to review our article. We hope these revisions are up to the journal’s publication standards. We believe that the updated manuscript has substantially improved with the reviewers’ comments and hope it will now be suitable for publication. Many thanks for your kind consideration.
This is a double-blind crossover trial assessing the effect of pre-exercise caffeine and glucose ingestion on endurance capacity in hypoxia. My comments are listed below:
- Introduction: authors must provide additional description on the target population (mountaineers etc.) for this research trial. This research caters to a very specific target population so enough background on this population must be included in this section.
RE: Thank you for your suggestion. We've added a sentence to describe the target population in the introduction (lines 82-85).
- Suggest authors to include a flow diagram/figure that provides readers with a quick outline of the research design including a brief description of the activities done at each visit.
RE: The experimental flow diagram has been added as suggested, as shown in (revised) Figure 1 (page 4).
- Results: Figure 1 is not mentioned the text and I feel Figure 1 can be presented as a Table.
RE: Thank you for your suggestion. The previous Figure 1 has been changed to Table 1 as suggested. (page 5).
- Discussion: authors must expand upon the value/significance of their research including details on the possible mechanism(s) of action(s). As currently written this section does not provide sufficient discussion of study results.
RE: Thank you for your suggestion. We have made some amendments to address the significance of the study and the possible mechanism in the Discussion (lines 269-271, 277-279).
General comments:
- please elaborate abbreviations at the first use in the manuscript.
RE: Thank you for your suggestion. We've rechecked the abbreviations in the manuscript and explained them in full at first mention.
- This manuscript needs professional English language assistance.
RE: Thank you. We have revised the manuscript with the assistance of a professional English editor.
- Authors should include a copy of their informed consent document as a supplemental file.
RE: We have forwarded a copy of their informed consent document to the editor before the review process.

Reviewer 3 Report
Comments and Suggestions for Authors
This great pilot study examines the effect of pre-workout carbohydrate load, which is normally taken post-workout. Although small-sized, the graphs are baffling. I have some remarks:
Discussion: caffeine intake yields some serious risks (incl. addiction), on which the authors should comment.
Statistics: normally, Bonferroni's correction is only applied in the case of at least 20 study subjects.
Comments on the Quality of English Language
Some grammatical errors throughout the text.
Author Response
This great pilot study examines the effect of pre-workout carbohydrate load, which is normally taken post-workout. Although small-sized, the graphs are baffling. I have some remarks:
RE: Thank you very much for taking the time to review our article. We hope these revisions are up to the journal’s publication standards. We believe that the updated manuscript has substantially improved with the reviewers’ comments and hope it will now be suitable for publication. Many thanks for your kind consideration.
Discussion: caffeine intake yields some serious risks (incl. addiction), on which the authors should comment.
RE: Thank you for the suggestion. The safety and efficacy of these caffeine doses have been documented in International Society of Sports Nutrition position stand (2021). We have addressed the issue in the revised manuscript (lines 160-163).
Statistics: normally, Bonferroni's correction is only applied in the case of at least 20 study subjects.
RE: Thanks for the comment. The data in this study have been first examined by the Shapiro-Wilk test method for the normal distribution, which shows that the data are normally distributed. According to Armstrong et al. (2015), it is stated that such a condition can be corrected using Bonferroni's correction [1]. Therefore, we adopted the statistical approach for this study. Please find the minor amendments in the statistics section (lines 177-178).
- Armstrong, R.A. When to use the Bonferroni correction. Ophthalmic and Physiological Optics 2014, 34, 502-508.

Round 2
Reviewer 1 Report
Comments and Suggestions for Authors
Thank you so much for your replies.
Comments on the Quality of English Language
No comment.